# Adversarial Style Mining for One-Shot Unsupervised Domain Adaptation

Yawei Luo[1,2,3],  Ping Liu[4,5],  Tao Guan[1] *,  Junqing Yu[1],  Yi Yang[2,4]

[1]School of Computer Science & Technology, Huazhong University of Science & Technology
[2]CCAI, Zhejiang University   [3]Baidu Research
[4]ReLER, University of Technology Sydney
[5]Institute of High Performance Computing, A*STAR, Singapore

## Abstract

We aim at the problem named One-Shot Unsupervised Domain Adaptation. Unlike traditional Unsupervised Domain Adaptation, it assumes that only one unlabeled target sample can be available when learning to adapt. This setting is realistic but more challenging, in which conventional adaptation approaches are prone to failure due to the scarce of unlabeled target data. To this end, we propose a novel Adversarial Style Mining approach, which combines the style transfer module and task-specific module into an adversarial manner. Specifically, the style transfer module iteratively searches for harder stylized images around the one-shot target sample according to the current learning state, leading the task model to explore the potential styles that are difficult to solve in the almost unseen target domain, thus boosting the adaptation performance in a data-scarce scenario. The adversarial learning framework makes the style transfer module and task-specific module benefit each other during the competition. Extensive experiments on both cross-domain classification and segmentation benchmarks verify that ASM achieves state-of-the-art adaptation performance under the challenging one-shot setting.

## 1   Introduction

Deep networks have significantly improved the performance for a wide variety of machine-learning problems and applications [32, 29]. Nevertheless, these impressive gains usually come with a price of massive amounts of manual labeled data. A popular trend in the current research community is to resort to simulated data, such as computer-generated scenes [36, 37], so that unlimited amount of automatic annotation is made available. However, this learning paradigm suffers from the shift in data distributions between the real and simulated domains, which poses a significant obstacle in adapting predictive models to the target task. The introduction of Domain Adaptation (DA) techniques aims to mitigate such performance drop when a trained agent encounters a different environment. By bridging the distribution gap between source and target domains, DA methods have shown their effect in many cross-domain tasks such as classification [27, 18], segmentation [19, 22, 23] and detection [3]. Although much progress has been made for domain adaptation, most of the previous efforts assume the availability of enough amounts of unlabeled target-domain samples. However, such an assumption can not always hold since not only data labeling but also data collection itself might be challenging, if not impossible, for the target task. In this data-scarce scenario with a limited amount of unlabeled data from target domains, most previous DA strategies, such as distribution

 Part of the work is conducted when Ping is in UTS. The code is publicly available at `https://github.com/RoyalVane/ASM`.

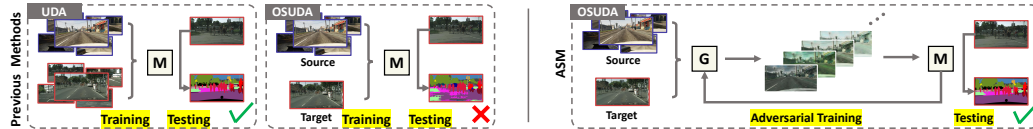

Figure 1: Conventional DA methods achieve good performance in UDA task but are prone to failure under the one-shot setting. We propose ASM to deal with such challenging data-scarce scenario.

alignment [12, 39], entropy minimization [41], or pseudo label generation [48, 45, 44], are all prone to failure. Consequently, design a specific algorithm for this realistic but more challenging learning scenario, *i.e.*, one-shot unsupervised domain adaptation (OSUDA), becomes necessary.

Some recent works [7, 25, 43, 42] aiming to vanilla UDA problems, try to learn style distribution in target domains based on the given unlabeled target data. The learned style distribution is utilized to translate source domain data to make them with a similar "appearance", *e.g.*, lighting, texture, etc., as target domain data. The model trained on stylized source data can naturally, hopefully, generalize well to the target domain. However, there are a few drawbacks if directly apply those vanilla style transfer (ST) methods in One-Shot UDA settings. First, both the style transfer module and the classifier could easily over-fit due to the scarce of target domain data. With only one target sample, it is hard to learn from it to catch the actual style distribution in the target domain. Second, in previous works [2, 4, 24, 12], ST and DA are usually carried out in a sequential, decoupled manner, which makes it hard for ST and DA benefit mutually to each other. That means, since the ST module can not get dynamic feedback from the classifier, it might produce inappropriate stylized samples, which might be either too "hard" or too "easy" for adapting the model in the current state.

To overcome these drawbacks above, in this paper, we propose the Adversarial Style Mining (ASM) algorithm effective for OSUDA [2]. As shown in Fig. 1, ASM is composed of a stylized image generator $G$ and a task-specific network $M$, *e.g.*, FCN for segmentation task. We design $G$ to generate arbitrary style from a sampling vector $\varepsilon$. We can change the style of the generated image by simply modifying $\varepsilon$. Unlike previous style translation works [46, 14], our $\varepsilon$ is initialized by the sole given target sample and will be updated according to the feedback from $M$ so as to match the learning ability of $M$ in a dynamic manner, while $M$ is trained to segment or classify the stylized images from $G$ correctly. By this way, we construct $G$ and $M$ as an end-to-end adversarial regime. Specifically, $\varepsilon$ and $M$ are iteratively updated during the ASM training. On the one hand, $G$ starts to generate stylized images from the initial $\varepsilon$ and constantly searches for harder styles for the current $M$. On the other hand, $M$ is trained based on these generated stylized images and returns its feedback, so $\varepsilon$ can be adjusted appropriately. In such an adversarial paradigm, we can efficiently produce stylized images that boost the domain adaptation, thus guiding $M$ to "see" more possible styles in the target domain beyond the solely given sample. Our main contributions are summarized as follows:

(1) We present an adversarial style mining (ASM) method to solve One-Shot Unsupervised Domain Adaptation (OSUDA) problems. ASM combines a style transfer module and a task-specific model into an adversarial manner, making them mutually benefit to each other during the learning process. ASM iteratively searches for new beneficial stylized images beyond the one-shot target sample, thus boosting the adaptation performance in data-scarce scenario.

(2) We propose a novel style transfer module, named Random AdaIN (RAIN), as a key component for achieving ASM. It makes the style searching a differentiable operation, hence enabling an end-to-end style searching using gradient back-propagation.

(3) We evaluate ASM on both cross-domain classification and cross-domain semantic segmentation in one-shot settings, showing that our proposed ASM consistently achieves superior performance over previous UDA and one-shot UDA approaches.

## 2 Related Work

### 2.1 Domain Adaptation

Based on the theory of Ben-David *et al.* [1], the majority of recent DA works [28] lay emphasis on how to minimize the domain divergence. Some methods [13, 26, 20] aim to align the latent feature

distribution between two domains, among which the most common strategy is to match the marginal distribution within the adversarial training framework [30, 40]. More similar to our method are approaches based on the image-to-image translation aiming to make images indistinguishable across domains, where an incomplete list of prior work includes [24, 43, 25, 10]. Joint consideration of image and feature level DA is studied in [12]. Besides alignment the latent features, Tsai *et al.* [39] found that directly aligning the output space is more effective in semantic segmentation. Based on the output space alignment, Vu *et al.* [41] further leveraged the entropy minimization to minimize the uncertainty of predictions in target data. Another popular branch is to extract confident target labels and use them to train the model explicitly [48, 47].

## 2.2 Style Transfer

Style transfer aims at altering the low-level visual style within an image while preserving its high-level semantic content. Gatys *et al.* [8] proposed the seminal idea to combine content loss and style loss based on the pre-trained neural networks on ImageNet [6]. Based on this pioneering work, Huang *et al.* [14] proposed the AdaIN to match the mean and variance statistics of the latent embedding of the content image and style image, then decoded the normalized feature into a stylized image. Another line of works [24, 46, 15] is based on the generative adversarial network (GAN) [11], which employs a discriminator to supervise the style transfer. Related to our settings, one-shot style transfer techniques have drawn more attention [2, 4] recently. More similar to our method are to use the style transfer as a data augmentation strategy [42, 17]. However, these works usually regard the style transfer as a single module and do not consider the interaction to other tasks.

# 3 Method

## 3.1 Problem Settings and Overall Idea

In the training process of OSUDA, we have access to the source data $X_S$ with labels $Y_S$, but only *one* unlabeled target data $x_T \in X_T$. The goal is to learn a model $M$ based on those data to correctly predict the labels for the target domain.

Overall, we suggest guiding the task-specific model $M$ to explore more possible styles in the target domain beyond the solely given sample. Introducing slight noise to the style from the *sole* given target sample is not an ideal solution in this case since it can not avoid overfitting. Purely randomly generating various styles, on the other hand, would produce images that are excessively deviated from the target distribution, which are either unrealistic or too hard for $M$. In our method, we propose to regard the style of the solely given target sample as an "anchor style". Starting from this anchor style, more stylized images are searched by a generator $G$ to boost the generalization of $M$. At the same time, the updated $M$ can provide a dynamic feedback for $G$ to determine the searching direction for new styles. Comparing to the old styles, those new styles are gradual "harder" for the $M$ to adapt and therefore provide stronger supervision to achieve a stronger $M$. In the following we will first introduce the design of $G$ dubbed Random AdaIN (RAIN). RAIN makes the style searching a differentiable operation, hence enabling an end-to-end adversarial style mining (ASM).

## 3.2 Random AdaIN

We first propose a module named Random Adaptive Instance Normalization (RAIN) as the stylized image generator $G$, which can easily respond to the feedback from the task-specific model $M$. RAIN equips the AdaIN [14] with a variational auto-encoder (named style VAE) in the latent space. For the AdaIN part, similar to [14], we employ the pre-trained VGG-19 as encoder $E$, to compute the loss function to train the decoder $D$:

$$\mathcal{L}_{Adain} = \mathcal{L}_c + \lambda_s \mathcal{L}_s, \tag{1}$$

where $\mathcal{L}_c$ and $\mathcal{L}_s$ denote content loss and style loss respectively, and $\lambda_s$ is a hyper-parameter controlling the relative importance of the two losses. AdaIN re-normalizes the features of content images $f_c$ to have the same channel-wise mean and standard deviation as the features of a selected style image $f_s$ as follows:

$$\text{AdaIN}(f_c, f_s) = \sigma(f_s) \left( \frac{f_c - \mu(f_c)}{\sigma(f_c)} \right) + \mu(f_s), \tag{2}$$

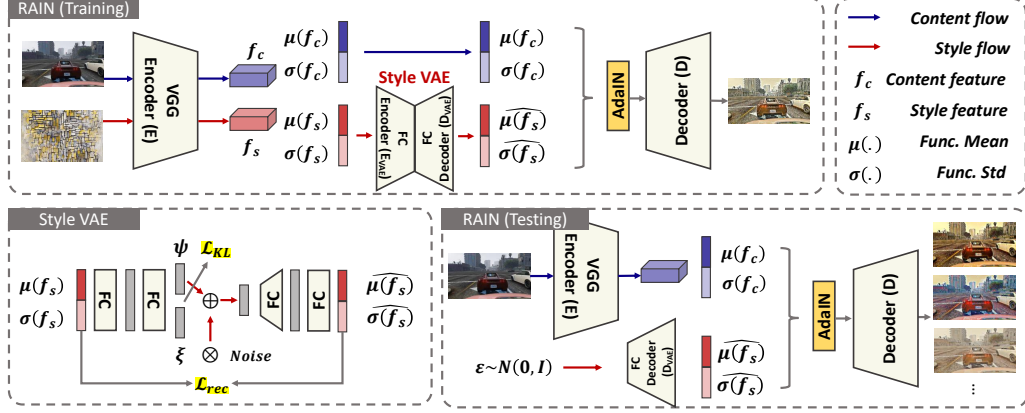

Figure 2: Overview of our proposed "Random AdaIN (RAIN)" module (See Top). Vanilla AdaIN regards each "style" as a pair of "mean $\mu(f_s)$" and "variation $\sigma(f_s)$" of the style image features $f_s$. Based on the vanilla AdaIN, we employ an extra VAE (See Left Bottom) in the latent space to encode the "style" (*i.e.*, $\mu(f_s)$ and $\sigma(f_s)$) into a standard distribution. In the testing stage (See Right Bottom), on the one hand, RAIN enables us to generate arbitrary new styles from some sampled vectors $\varepsilon$, without the need for style images. On the other hand, we can simply generate **other reasonable styles** near $\varepsilon$ by passing a small perturbation to $\varepsilon$. These properties of RAIN make the style searching a differentiable operation, hence enabling an end-to-end style mining using gradient back-propagation, which is the key to achieve ASM.

where $\mu(.)$ and $\sigma(.)$ denote channel-wise mean and standard deviation operations, respectively.

The style VAE, as shown in Fig. 2, is composed of an encoder $E_{vae}$ and a decoder $D_{vae}$, both of which contain two FC layers. $E_{vae}$ encodes $\mu(f_s) \odot \sigma(f_s)$ (where $\odot$ denotes "concatenate") to a Gaussian distribution $N(\psi, \xi)^3$, and $D_{vae}$ decodes a sampling $\varepsilon$ from such distribution aiming to reconstruct the original style. Therefore, besides the conventional training scheme for AdaIN, we have two extra losses for training the Style VAE. The overall training objective for RAIN is to minimize the following loss:

$$\mathcal{L}_{RAIN} = \mathcal{L}_c + \lambda_s \mathcal{L}_s + \lambda_k \mathcal{L}_{KL} + \lambda_r \mathcal{L}_{Rec} \tag{3}$$

Within Eq. 3, the latter two terms form the training loss for Style VAE:

$$\mathcal{L}_{KL} = \text{KL}[\mathcal{N}(\psi, \xi) || \mathcal{N}(0, I)] \tag{4}$$

$$\mathcal{L}_{Rec} = \|\mu(f_s) \odot \sigma(f_s), \mu(\widehat{f_s) \odot \sigma}(f_s)\|_2 \tag{5}$$

where $\mu(\widehat{f_s) \odot \sigma}(f_s)$ denotes the reconstructed style vector from a sampling $\varepsilon \sim \mathcal{N}(\psi, \xi)$.

### 3.3 Adversarial Style Mining

We illustrate ASM framework in Fig. 3 and the corresponding pipeline in Alg. 1. We employ a pre-trained RAIN module as the stylized image generator $G = \{E, D, E_{vae}, D_{vae}\}$, whose parameters are kept fixed during the training. The generated images will be forwarded to the task model $M$, with the goal of leading $M$ to generalize to the target domain. Given one-shot target sample $x_T$, we can first obtain a latent distribution $\mathcal{N}(\psi, \xi)$, where $\psi, \xi = E_{vae}(E(x_T))$. Each time we are given a source domain image $x_S$, we can sample $\varepsilon$ from $\mathcal{N}(\psi, \xi)$ and decode it to an initial style vector $\mu(\widehat{f_s) \odot \sigma}(f_s) = D_{vae}(\varepsilon)$, from which we can further generate an initial stylized image $x_1$. Since the current style of $x_1$ is very close to $x_T$, we can regard it as an anchor-style image.

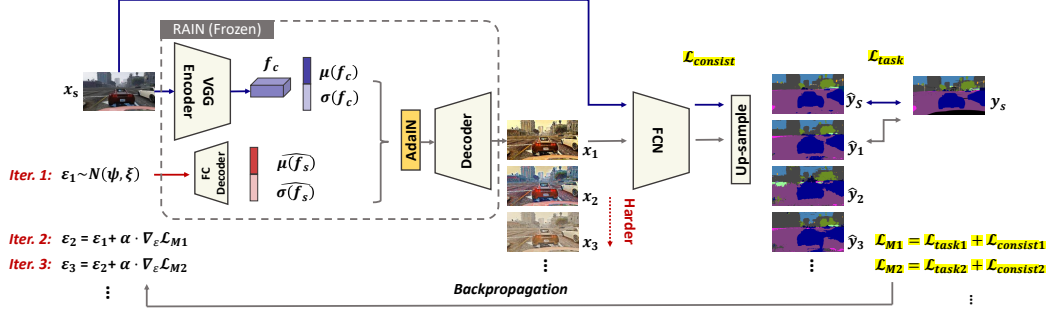

Figure 3: The framework of ASM. It consists of a stylized image generator $(G)$ and a task-specific network $(M)$. Here we take the semantic segmentation task as an example, where $M$ can be any FCN-based structure. $G$ is a pre-trained RAIN module described in section 3.2. By updating its input $\varepsilon$ based on the feedback from $M$, $G$ can continually generate harder samples for $M$ to boost its generality. First, we sample an initial style vector $\varepsilon_1$ from a Gaussian distribution. In our one-shot scenario, such Gaussian distribution is defined by $\psi$ and $\xi$, which is extracted from the one-shot target image $x_t$. Second, a source domain image $x_s$, together with $\varepsilon_1$, are forwarded to RAIN to generate a stylized image $x_1$, which is then fed into $G$ to produce the training loss $\mathcal{L}_{M1}$. We minimize $\mathcal{L}_{M1}$ to train $M$ to better generalize to $x_1$, and also prepare for next iteration by searching for the new vector $\varepsilon_i$ around $\varepsilon_1$ that can generate harder stylized image. Specifically, we update $\varepsilon_1$ by adding a small perturbation whose direction equals to the elements of the gradient of the loss function with respect to $\varepsilon_1$. Finally, the $\varepsilon_i$, as a harder sampling mined by ASM, will be fed into the pipeline the same way as $\varepsilon_1$ to bootstrap next iteration.

---

**Algorithm 1:** Adversarial Style Mining

**Input:** Source domain data $X_S$; source domain label $Y_S$; one-shot target domain data $x_T \in X_T$; a pre-trained RAIN module $G = \{E, D, E_{vae}, D_{vae}\}$; task model $M$ with parameter $\theta$; learning rate $\alpha$, $\beta$; max searching depth $n$

**Output:** Optimal $\theta^*$

1  Randomly initialize $\theta$;
2  $\psi, \xi = E_{vae}(E(x_T))$;
3  **for** $x_S \in X_S$ **do**
4  $\quad$ $f_c = E(x_S)$;
5  $\quad$ Sampling $\varepsilon \sim \mathcal{N}(\psi, \xi)$;
6  $\quad$ **for** $i = 1, ..., n$ **do**
7  $\quad\quad$ Reconstruct the style vector: $\widehat{\mu(f_s)} \odot \widehat{\sigma(f_s)} = D_{vae}(\varepsilon)$;
8  $\quad\quad$ Generate stylized image $x_{style} = D(\text{AdaIN}(f_c, \mu(f_c), \sigma(f_c), \widehat{\mu(f_s)}, \widehat{\sigma(f_s)}))$;
9  $\quad\quad$ Update model parameters: $\theta \leftarrow \theta - \alpha \nabla_\theta \mathcal{L}_M(M(x_{style}), y_S)$;
10 $\quad\quad$ Update sampling: $\varepsilon \leftarrow \varepsilon + \beta \nabla_\varepsilon \mathcal{L}_M(M(x_{style}), y_S)$;
11 **return** $\theta$ as $\theta^*$;

---

Following the overall idea in Sec. 3.1, the next step is to search for some new styles around anchor style iteratively. We construct this step as an adversarial paradigm. On the one hand, we update task model $M$ by minimizing a cost function $\mathcal{L}_M$, training $M$ to classify (or segment) $x_1$ rightly. On the other hand, we update $\varepsilon$ by adding a small perturbation whose direction is consistent with the gradient of the cost function with respect to $\varepsilon$. In this adversarial spirit, $M$ could learn to handle those more arduous samples $x_i$ around the one-shot anchor rightly, thus performing better on the unseen target domain samples.

### 3.4 Cost Function

Two losses are used to train the task model $M$.

**Task Loss.** We employ the task loss to train $M$ to learn knowledge from source label:

$$\mathcal{L}_{task} = \ell(M(x_S), y_S) , \tag{6}$$

where $x_S$ can be original or stylized source data. $\ell(\cdot, \cdot)$ is a task-specific cost function, *e.g.,* multi-class cross entropy for segmentation task.

**Consistency Loss.** To further encourage task model $M$ to distill the domain invariant feature, we employ a consistency loss as follows:

$$\mathcal{L}_{consist} = \frac{\sum_{i=1}^{N} \|z - \overline{z}\|_2}{N} , \tag{7}$$

where $z$ denotes the latent features from the second last layer of $M$, $\overline{z}$ denotes the average value of $z$ across a $N$-sized batch. The motivation behind is that a source image under different stylization should maintain similar semantic information in deep layers. Such loss constrains the semantic consistency across a mini-batch of images, which have same content but different styles.

Then the overall cost function to train $M$ is:

$$\mathcal{L}_M = \mathcal{L}_{task} + \lambda \mathcal{L}_{consist}, \tag{8}$$

where $\lambda$ denotes a hyper-parameter controlling the relative importance of the two losses.

## 4 Experiments

### 4.1 Experimental Settings

We evaluate ASM together with several state-of-the-art UDA algorithms on both classification and segmentation tasks using PaddlePaddle and Pytorch. We use MNIST [21]-USPS [16]-SVHN [35] benchmarks to evaluate ASM on one-shot cross domain classification task. For one-shot cross-domain segmentation task, we evaluate ASM on two benchmarks, *i.e.*, SYNTHIA [37] → Cityscapes [5] and GTA5 [36] → Cityscapes. We run each OSUDA experiment for 5 times to get the average result, where each time we randomly select one-shot sample from the target domain.

Besides the task-specific datasets, we extra leverage some data as the "style images" to train the RAIN. Here we follow [14] to use a dataset of paintings mostly collected from WikiArt. However, there is no limit to choose any other website data since no annotation is required for style images. More details on experimental settings are given in **Appendix A** and **B**.

### 4.2 Image Classification

In this experiment, we evaluate the adaptation scenario across MNIST-USPS-SVHN datasets. We present the adaptation results on task $M \rightarrow S$, $U \rightarrow S$ and $M \rightarrow U$ in Table 1 with comparisons to the state-of-the-art domain adaptation methods. We also implement several classic style transfer methods such as CycleGAN [46] and MUNIT [15] under the one-shot setting. From the table, we can observe that on the task $M \rightarrow S$ and $U \rightarrow S$, ASM produces the state-of-the-art classification accuracy (46.3% and 40.3%), significantly outperforming other competitors under one-shot UDA settings. Moreover, ASM performs even better than the few-shot supervised methods, indicating our strategy can make the utmost of the given one-shot sample. To make our analysis more convincing, we visualize the learned representations in $M \rightarrow S$ task via t-distributed stochastic neighbor embedding (t-SNE) [33] in Fig. 4. Nevertheless, we can also find that all the style transfer-based methods, including CycleGAN, MUNIT, and ASM, fall short on the $M \rightarrow U$ case. Such a result is reasonable since the domain shift between $M$ and $U$ lies in the content itself but not in the style difference. This phenomenon reveals the cases that ASM and other style transfer-based methods are not applicable.

### 4.3 Semantic Segmentation

For the cross-domain segmentation task, we compare our method with several recent UDA methods, including **CBST** [48], **AdaptSeg** [39], **CLAN** [31], **ADVENT** [41]. Divided by the different strategies, these methods can be categorized into three groups: (i) alignment-based method, *i.e.*, AdaptSeg, CLAN, whose idea is to make the distribution of two domains to be similar; (ii) Entropy

Table 1: Cross-domain classification on MNIST-USPS-SVHN (M-U-S) datasets. $L/UL$ denotes the labeled / unlabeled data used in training. $\#TS$ denotes the number of target sample.

| $Method$ | $\#TS$ | $L/UL$ | $M \rightarrow S$ | $U \rightarrow S$ | $M \rightarrow U$ |
|---|---|---|---|---|---|
| UDA | | | | | |
| Source Only | - | - | 20.3 | 15.3 | 65.4 |
| DRCN [9] | $all$ | $UL$ | 40.1 | - | 91.8 |
| GenToAdapt [38] | $all$ | $UL$ | 36.4 | - | 92.5 |
| Few-shot UDA | | | | | |
| FADA [34] | $10(1/class)$ | $L$ | 37.7 | 27.5 | 85.0 |
| FADA [34] | $50(5/class)$ | $L$ | 46.1 | 37.9 | 92.4 |
| One-shot UDA | | | | | |
| CycleGAN [46] | 1 | $UL$ | 28.2 | 20.7 | 66.8 |
| MUNIT [15] | 1 | $UL$ | 35.0 | 26.5 | 67.4 |
| OST [2] | 1 | $UL$ | 42.5 | 34.0 | **74.8** |
| ASM (Ours) | 1 | $UL$ | **46.3** | **40.3** | 68.0 |

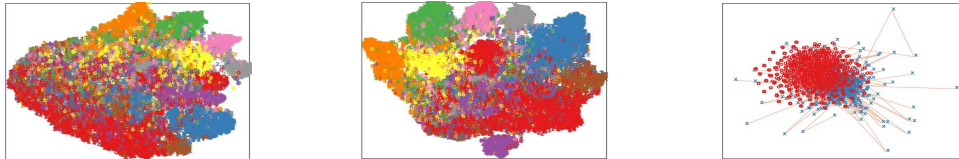

Figure 4: The t-SNE [33] visualization in $M \rightarrow S$ task. **Left & Middle:** The feature distribution learned by OST [2] and ASM (different class is shown in different color). **Right**: The style distribution of real target domain $S$ (Red) and searched by ASM (Blue). Yellow lines represent the search paths.

minimization-based method, *i.e.*, ADVENT, which tends to minimize the uncertainty of predictions in target data; and (iii) Pseudo label-based method, *i.e.*, CBST, which extracts confident target labels and use them to train the model explicitly. For a clear comparison, we also report the segmentation result when using the source data only or using all the labeled target data to train the model. As we can observe, there is a large performance gap ($36.6\%$ vs $70.4\%$) between the two approaches.

Firstly, we evaluate these methods under the conventional UDA settings that all the unlabeled target data are available. As shown in Table 2, these conventional UDA strategies can give a huge boost to the source-only baseline, bringing at least $5\%$ improvement in terms of mIoU. In such a data-rich scenario, ASM can extract variant anchor styles $\{\psi_i, \xi_i\}$ to exploit more possible styles in the target domain. Accordingly, ASM yields $45.5\%$ in terms of mean IOU, which is on par with other methods and slightly better than it performs under the one-shot setting. Such results indicate that ASM can also be applied for UDA problems.

Secondly, we compare ASM with the above methods under the OSUDA setting. Not surprisingly, all the competitors deteriorate significantly in such a data-scarce scenario. Some of them even yield worse mIoU than the source only baseline due to the overfitting to the One-shot target sample. Besides the UDA methods, we also compare our method with state-of-the-art one-shot style transfer methods, *e.g.,* OST [2] and CycleGAN [46]. To fairly compare these methods with ASM, we additionally train a ResNet-101-based segmentor upon the generated samples from OST or CycleGAN. We find that both CycleGAN and OST can improve the mIoU over the source only baseline, proofing that the style transfer is a robust strategy facing the data-scarce scenario. Furthermore, ASM boosts the mIoU to a new benchmark of $44.5\%$, which demonstrates the advantage of our adversarial scheme in ASM over the sequential combination of style transfer and segmentation like OST and CycleGAN.

Finally, by comparing the performance of ASM under UDA and OSUDA settings, we can observe that reducing the visible target data does not hurt ASM ($45.5\% \rightarrow 44.5\%$) as much as it hurts the other competing methods ($\sim 44.0\% \rightarrow \sim 37.0\%$). The smaller performance drop between One-Shot and conventional settings further proves that ASM can efficiently search for useful styles from the solely given samples. Such a self-mining mechanism minimizes the impact of missing target data. The same observation can be also found in SYNTHIA $\rightarrow$ Cityscapes task (See **Appendix C**).

Table 2: Adaptation from GTA5 [36] to Cityscapes [5]. We present per-class IoU and mean IoU. "A", "E" and "P" represent three lines of method, *i.e.,* Alignment- , Entropy minimization- and Pseudo label-based DA. #TS denotes the number of target sample used in training. *Gain* indicates the mIoU improvement over using the source only.

| | Meth. | #TS | road | side. | buil. | wall | fence | pole | light | sign | vege. | terr. | sky | pers. | rider | car | truck | bus | train | motor | bike | **mIoU** | **gain** |
|---|---|---|---|---|---|---|---|---|---|---|---|---|---|---|---|---|---|---|---|---|---|---|---|
| Source only | — | — | 75.8 | 16.8 | 77.2 | 12.5 | 21.0 | 25.5 | 30.1 | 20.1 | 81.3 | 24.6 | 70.3 | 53.8 | 26.4 | 49.9 | 17.2 | 25.9 | 6.5 | 25.3 | 36.0 | 36.6 | — |
| Fully supervised | — | — | 97.9 | 81.3 | 90.3 | 48.8 | 47.4 | 49.6 | 57.9 | 67.3 | 91.9 | 69.4 | 94.2 | 79.8 | 59.8 | 93.7 | 56.5 | 67.5 | 57.5 | 57.7 | 68.8 | 70.4 | 33.8 |
| *UDA* | | | | | | | | | | | | | | | | | | | | | | | |
| CycleGAN [46] | A | All | 81.7 | 27.0 | 81.7 | 30.3 | 12.2 | 28.2 | 25.5 | 27.4 | 82.2 | 27.0 | 77.0 | 55.9 | 20.5 | 82.8 | 30.8 | 38.4 | 0.0 | 18.8 | 32.3 | 41.0 | 4.4 |
| AdaptSeg [39] | A | All | 86.5 | 36.0 | 79.9 | 23.4 | 23.3 | 23.9 | 35.2 | 14.8 | 83.4 | 33.3 | 75.6 | 58.5 | 27.6 | 73.7 | 32.5 | 35.4 | 3.9 | 30.1 | 28.1 | 42.4 | 5.8 |
| CLAN [31] | A | All | 87.0 | 27.1 | 79.6 | 27.3 | 23.3 | 28.3 | 35.5 | 24.2 | 83.6 | 27.4 | 74.2 | 58.6 | 28.0 | 76.2 | 33.1 | 36.7 | 6.7 | 31.9 | 31.4 | 43.2 | 6.6 |
| Advent [41] | A+E | All | 89.4 | 33.1 | 81.0 | 26.6 | 26.8 | 27.2 | 33.5 | 24.7 | 83.9 | 36.7 | 78.8 | 58.7 | 30.5 | 84.8 | 38.5 | 44.5 | 1.7 | 31.6 | 32.4 | 45.5 | 8.9 |
| CBST [48] | P | All | 86.8 | 46.7 | 76.9 | 26.3 | 24.8 | 42.0 | 46.0 | 38.6 | 80.7 | 15.7 | 48.0 | 57.3 | 27.9 | 78.2 | 24.5 | 49.6 | 17.7 | 25.5 | 45.1 | 45.2 | 8.6 |
| ASM (Ours) | A | All | 89.8 | 38.2 | 77.8 | 25.5 | 28.6 | 24.9 | 31.2 | 24.5 | 83.1 | 36.0 | 82.3 | 55.7 | 28.0 | 84.5 | 45.9 | 44.7 | 5.3 | 26.4 | 31.3 | 45.5 | 8.9 |
| *One-shot UDA* | | | | | | | | | | | | | | | | | | | | | | | |
| CycleGAN [46] | A | One | 80.3 | 23.8 | 76.7 | 17.3 | 18.2 | 18.1 | 21.3 | 17.5 | 81.5 | 40.1 | 74.0 | 56.2 | **38.3** | 77.1 | 30.3 | 27.6 | 1.7 | **30.0** | 22.2 | 39.6 | 3.0 |
| AdaptSeg [39] | A | One | 77.7 | 19.2 | 75.5 | 11.7 | 6.4 | 16.8 | 18.2 | 15.4 | 77.1 | 34.0 | 68.5 | 55.3 | 30.9 | 74.5 | 23.7 | 28.3 | **2.9** | 14.4 | 18.9 | 35.2 | -1.4 |
| CLAN [31] | A | One | 77.1 | 22.7 | 78.6 | 17.0 | 14.8 | 20.5 | 23.8 | 12.0 | 80.2 | 39.5 | 74.3 | 56.6 | 25.2 | 78.1 | 29.3 | 31.2 | 0.0 | 19.4 | 16.7 | 37.7 | 1.1 |
| Advent [41] | A+E | One | 76.1 | 15.1 | 76.6 | 14.4 | 10.8 | 17.5 | 19.8 | 12.0 | 79.2 | 39.5 | 71.3 | 55.7 | 25.2 | 76.7 | 28.3 | 30.5 | 0.0 | 23.6 | 14.4 | 36.1 | -0.5 |
| CBST [48] | P | One | 76.1 | 22.2 | 73.5 | 13.8 | 18.8 | 19.1 | 20.7 | 18.6 | 79.5 | 41.3 | 74.8 | 57.4 | 19.9 | 78.7 | 21.3 | 28.5 | 0.0 | 28.0 | 13.2 | 37.1 | 0.5 |
| OST [2] | A | One | 84.3 | 27.6 | 80.9 | 24.1 | 23.4 | 26.7 | 23.2 | 19.4 | 80.2 | **42.0** | **80.7** | **59.2** | 20.3 | 84.1 | 35.1 | 39.6 | 1.0 | 29.1 | 23.2 | 42.3 | 5.7 |
| ASM (Ours) | A | One | **86.2** | **35.2** | **81.4** | **24.2** | **25.5** | **31.5** | **31.5** | **21.9** | **82.9** | 30.5 | 80.1 | 57.3 | 22.9 | **85.3** | **43.7** | **44.9** | 0.0 | 26.5 | **34.9** | **44.5** | **7.9** |

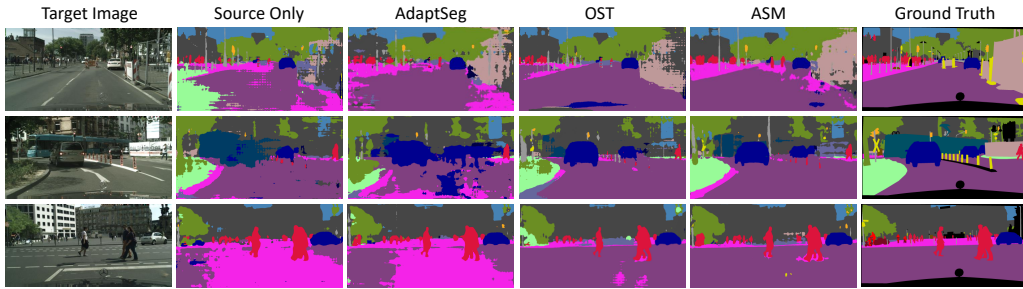

Figure 5: Qualitative results of OSUDA segmentation on GTA5 → Citys. For each target image, we show the source only result, adapted result with AdaptSeg [39], OST [2], ASM and the ground truth.

## 4.4 Analysis of the proposed method

**Ablation Study.** In this section we conduct an ablation study on the consistency loss described in Eq. 7. It is employed to encourage the model $M$ to distill similar deep features when encountering the same image under different stylization. The results are given in Table 3. As we can see, imposing a consistency loss to ASM can boost both classification (+1.5%) and segmentation (+0.5%) tasks. In fact, the consistency loss and the task loss are complementary since the former aims to supervise the latent features while the later is on the output. Other ablation studies are given in **Appendix C**.

**Style Distribution.** Here we analyze the distribution of new styles explored by ASM. Obviously, we hope that the new samples searched by ASM can overlap the real style distribution in the target domain. However, it is nearly impossible under the one shot setting since only one target sample is available during training. Here we visualize the embedded styles in $M \to S$ task via t-SNE [33] (See Fig. 4 Right.), where the red points denote the real target styles while the blue ones represent the styles mined by ASM. We can observe that ASM can efficiently search for "unseen" styles around the anchor style, thus promoting domain alignment in terms of style.

**Variant Study on Different Sampling Strategy.** In this section we conduct the variation study on the $\varepsilon$ sampling methods. Based on the proposed RAIN module, we consider three different sampling strategies: (a) anchored sampling, *i.e.*, $\varepsilon_i \sim \mathcal{N}(\psi, \xi)$; (b) random sampling, *i.e.*, $\varepsilon_i \sim \mathcal{N}(0, I)$; and (c) ASM. The visualization comparison of the three sampling variants is depicted in Fig. 6. We also report the mIoU using these three strategies on task GTA5 → Cityscapes in Table 4. As shown first row in Fig. 6, anchored sampling would lead to very similar images near the given target sample. On

Table 3: Ablation study on consistency loss (CL).

| M→S w/ CL | M→S w/o CL | GTA5 → C. w/ CL | GTA5 → C. w/o CL |
|:---:|:---:|:---:|:---:|
| **46.3** | 44.8 (1.5↓) | **44.5** | 44.0 (0.5↓) |

Table 4: Segmentation performance on task GTA5 → Cityscapes, using variant of sampling strategy of $\varepsilon$.

| **Sampling** | Anchored | Random | ASM |
|:---:|:---:|:---:|:---:|
| **mIoU** | 42.9 | 42.4 | **44.5** |

the other hand, random sampling would produce many styles that are not helpful for the adaptation (See second row). Finally, the last row shows the stylized images found by ASM. From left to right, the generated style is increasingly different from the anchor style and harder for $M$. Together with the fact that ASM outperforms the former two sampling strategies by around $2\%$ in terms of mIoU, we can conclude that ASM offers better sampling strategy for the one-shot adaptation scenario.

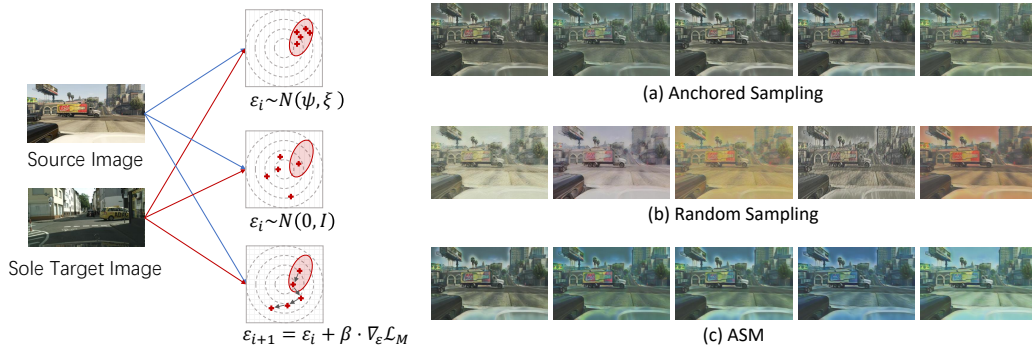

Figure 6: The comparison of different sampling strategies, where we visualize their respective generated images.

## 5 Conclusion

In this paper, we introduce the Adversarial Style Mining (ASM) approach, aiming at the unsupervised domain adaptation (UDA) problem in case of a target-data-scarce scenario. ASM combines the style transfer module and the task model in an adversarial manner, iteratively and efficiently searching for new stylized samples to help the task model to adapt to the almost unseen target domain. ASM is general in the sense that the task-specific sub-network $M$ can be changed according to different cross-domain tasks. Experimental results on both classification and segmentation tasks validate the effectiveness of ASM, which yields state-of-the-art performance compared with other domain adaptation approaches in the one-shot scenario. The limitation of ASM is the assumption that the distribution gap between source and target domain consists in the style difference. We leave it as the future work to relax such assumption and build a stronger model to cope with general domain gaps.

## Broader Impact

Using as few manual annotations as possible to get a model that can handle data in different environments and states has been a long standing topic in computer vision and artificial intelligence community. UDA has been dedicating to this topic while our proposed ASM has taken a step further by relaxing the assumption of sufficient target data. In many scenarios, not only data labeling but also data collection itself might be challenging, if not impossible, for the target task. For example, it could be hard to acquire rare disease information with privacy or to shoot videos under extreme weather conditions. From this point of view, the proposed ASM has the potential to not only reduce the amount of labor required to collect data, but also to protect private data from being leaked. Overall, the impact of our proposed ASM to the research community is positive and beneficial.

## Acknowledgments and Disclosure of Funding

This work is supported by National Key R&D Program of China under Grant No. 2020AAA0108800 and National Key R&D program under Grant No. 2016YFB1000204. This work is also partially supported by CCF-Baidu Open Fund under Grant No. CCF-BAIDU OF2020016. Most importantly, I'd like to express my deepest gratitude to my wife, Dan Zhou. The paper was written during her pregnancy. Many thanks to her understanding and support. Now I am looking forward to the birth of our lovely baby, just like I was expecting this paper to be accepted.

## Footnotes

*Corresponding author (qd_gt@hust.edu.cn).

[2]It should be noted that ASM can also be applied for UDA problems.

[3]We use the notation $\psi$ for mean and $\xi$ for standard deviation in Style VAE, in order to avoid confusion to the $\mu(.)$ and $\sigma(.)$ in AdaIN.

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
