[Supplementary Material]

# Adversarial Style Mining for One-Shot Unsupervised Domain Adaptation (Appendix)

**Yawei Luo**[1,2,3], **Ping Liu**[4,5], **Tao Guan**[1]\*, **Junqing Yu**[1], **Yi Yang**[2,4]

[1]School of Computer Science & Technology, Huazhong University of Science & Technology
[2]CCAI, Zhejiang University   [3]Baidu Research
[4]ReLER, University of Technology Sydney
[5]Institute of High Performance Computing, A\*STAR, Singapore

## A. Datasets Details and Evaluation Protocols

We evaluate ASM together with several state-of-the-art UDA algorithms on both classification and segmentation tasks. We use MNIST [8]-USPS [7]-SVHN [12] benchmarks to evaluate ASM on one-shot cross domain classification task, where MNIST (M) and USPS (U) contain images of hand-writing digits from 0 to 9 while SVHN (S) captures some images of the house number in the wild. We select three adaptation tasks, *i.e.*, $M \to S$, $U \to S$ and $M \to U$, to evaluate ASM. Following the experimental setting in [17, 9, 11], we use all the source domain data in the first two tasks while randomly selecting 2,000 images from MNIST in task $M \to U$. We use the classification accuracy as the evaluation metric.

For one-shot cross-domain segmentation task, we evaluate ASM on two benchmarks, *i.e.*, SYNTHIA [15] $\to$ Cityscapes [5] and GTA5 [14] $\to$ Cityscapes. Cityscapes is a real-world dataset with 5,000 street scenes which are divided into a training set with 2,975 images, a validation set with 500 images and a testing set with 1,525 images. We use Cityscapes as the one-shot target domain. GTA5 contains 24,966 high-resolution images, automatically annotated into 19 classes. The dataset is rendered from a modern computer game, Grand Theft Auto V, with labels fully compatible with those of Cityscapes. SYNTHIA contains 9,400 synthetic images compatible with the Cityscapes annotated classes. We use SYNTHIA or GTA5 as the source domain in evaluation. In terms of the evaluation metrics, we leverage Insertion over Union (IoU) to measure the performance of the compared methods.

## B. Experimental Setup Details

We use PyTorch [13] as well as PaddlePaddle for our implementation, both achieving similar performance. The training process is composed of two stages. In the first stage, we use source images and style images to train the RAIN module. In the second stage, we fix RAIN and train the task model within the ASM framework. For **classification task**, we employ ResNet-18 [6] as the backbone and SGD [3] as the optimizer, with a weight decay of $5e$-4. We train the network for a total of $30k$ iterations, with the first 600 as the warm-up stage [1] during which the learning rate increases linearly from 0 to the initial value. Then the learning rate is divided by ten at 10k and 20k iterations. We resize the input images to $64 \times 64$ and the batch size is set to 64. For **segmentation task**, we leverage ResNet-101 [6]-based DeepLab-v2 [4] as the backbone of segmentor. To reduce the memory footprint, we resize the original image to $1,280 \times 720$ and random crop $960 \times 480$ as the input. We use SGD [3] with a momentum of 0.9 and a weight decay of $5e$-4 as the optimizer. The initial

Part of the work is conducted when Ping is in UTS. The code is publicly available at `https://github.com/RoyalVane/ASM`.

Table 1: Adaptation from SYNTHIA [15] to Cityscapes [5]. We present per-class IoU and mean IoU for evaluation. ASM and state-of-the-art domain adaptation methods are compared.

| | Meth. | #TS | road | side. | buil. | light | sign | vege. | sky | pers. | rider | car | bus | motor | bike | mIoU | gain |
|---|---|---|---|---|---|---|---|---|---|---|---|---|---|---|---|---|---|
| **SYNTHIA → Cityscapes** | | | | | | | | | | | | | | | | | |
| Source only | — | — | 55.6 | 23.8 | 74.6 | 6.1 | 12.1 | 74.8 | 79.0 | 55.3 | 19.1 | 39.6 | 23.3 | 13.7 | 25.0 | 38.6 | — |
| Fully supervised | — | — | 95.1 | 72.9 | 87.3 | 46.7 | 57.2 | 87.1 | 92.1 | 74.2 | 35.0 | 92.1 | 49.3 | 53.2 | 68.8 | 70.1 | 31.5 |
| **UDA** | | | | | | | | | | | | | | | | | |
| AdaptSeg [16] | A | All | 84.3 | 42.7 | 77.5 | 4.7 | 7.0 | 77.9 | 82.5 | 54.3 | 21.0 | 72.3 | 32.2 | 18.9 | 32.3 | 46.7 | 8.1 |
| CLAN [10] | A | All | 81.3 | 37.0 | 80.1 | 16.1 | 13.7 | 78.2 | 81.5 | 53.4 | 21.2 | 73.0 | 32.9 | 22.6 | 30.7 | 47.8 | 9.2 |
| ADVENT [18] | A+E | All | 85.6 | 42.2 | 79.7 | 5.4 | 8.1 | 80.4 | 84.1 | 57.9 | 23.8 | 73.3 | 36.4 | 14.2 | 33.0 | 48.0 | 9.4 |
| CBST [20] | P | All | 53.6 | 23.7 | 75.0 | 23.5 | 26.3 | 84.8 | 74.7 | 67.2 | 17.5 | 84.5 | 28.4 | 15.2 | 55.8 | 48.4 | 9.8 |
| **One-shot UDA** | | | | | | | | | | | | | | | | | |
| AdaptSeg [16] | A | One | 64.1 | 25.6 | 75.3 | 4.7 | 2.7 | 77.0 | 70.0 | 52.2 | 20.6 | 51.3 | 22.4 | 19.9 | 22.3 | 39.1 | 0.5 |
| CLAN [10] | A | One | 68.3 | 26.9 | 72.2 | 5.1 | 5.3 | 75.9 | 71.4 | 54.8 | 18.4 | 65.3 | 19.2 | 22.1 | 20.7 | 40.4 | 1.8 |
| ADVENT [18] | A+E | One | 65.7 | 22.3 | 69.2 | 2.9 | 3.3 | 76.9 | 69.2 | 55.4 | 21.4 | 77.3 | 17.4 | 21.4 | 16.7 | 39.9 | 1.3 |
| CBST [20] | P | One | 59.6 | 24.1 | 72.9 | 5.5 | 13.8 | 72.2 | 69.8 | 55.3 | 21.1 | 57.1 | 17.4 | 13.8 | 18.5 | 38.5 | -0.1 |
| OST [2] | A | One | **75.3** | **31.6** | 72.1 | **12.3** | 9.3 | 76.1 | 71.1 | 51.1 | 17.7 | 68.9 | 19.0 | **26.3** | **25.4** | 42.8 | 4.7 |
| ASM (Ours) | A | One | 73.5 | 29.0 | **75.2** | 10.9 | **10.1** | 78.1 | 73.2 | 56.0 | 23.7 | 76.9 | 23.3 | 24.7 | 18.2 | **44.1** | **6.0** |

learning rates for SGD is set to $2.5e\text{-}4$ and is decayed by a poly policy, where the initial learning rate is multiplied by $(1 - \frac{iter}{max\_iter})^{power}$ with $power = 0.9$. We train the network for a total of $100k$ iterations, with the first $5k$ as the warm-up stage like in classification task. In our best model, we set hyper-parameters $\lambda = 2e-4$, $\lambda_s = 1.0$, $\lambda_k = 1.0$, $\lambda_r = 5.0$, respectively. The searching depth $n$ in each iteration is set to 5 in classification task and 2 in segmentation task. Although the inner loop runs $n$ times within each training step, we decreased the training epochs (outer loop) in order to guarantee that ASM has a similar number of back-propagation with the baseline. It not only ensures a fair comparison but also reduces the computational overhead.

## C. Additional Experimental Results

**Result on SYNTHIA → Cityscapes task.** We compare our method with several recent UDA and OSUDA methods, including **CBST** [20], **AdaptSeg** [16], **CLAN** [10], **ADVENT** [18], **OST** [2] and **CycleGAN** [19]. For a clear comparison, we also report the segmentation result when using the source data only or using all the labeled target data to train the model. As we can observe, there is a large performance gap ($38.6\%$ vs $70.1\%$) between the two approaches.

We evaluate these methods under the conventional UDA settings that all the unlabeled target data are available. As shown in Table 1, these conventional UDA strategies can give a huge boost to the source-only baseline, bringing at least $8\%$ improvement in terms of mIoU. However, when testing under the One-shot UDA setting, all the competitors deteriorate significantly in such a data-scarce scenario. Some of them even yield worse mIoU than the source only baseline due to the overfitting to the One-shot target sample. Besides the UDA methods, we also compare our method with state-of-the-art one-shot style transfer methods, *e.g.,* OST [2] and CycleGAN [19]. To fairly compare these methods with ASM, we additionally train a ResNet-101-based segmentor upon the generated samples from OST or CycleGAN. We find that both CycleGAN and OST can improve the mIoU over the source only baseline, proofing that the style transfer is a robust strategy facing the data-scarce scenario. Furthermore, ASM boosts the mIoU to a new benchmark of $44.1\%$, which demonstrates the advantage of our adversarial scheme in ASM over the sequential combination of style transfer and segmentation like OST and CycleGAN.

**Variant Study on Different Search Depth.** The search depth $n$ is a key hyper-parameter in ASM training process. In this variant experiment we test our model using a varying $n$ over a range {1, 5, 10, 20} for the classification task and a varying $n$ over a range {1, 2, 3, 4} for the segmentation task. Since the inner loop runs $n$ times within each training step, to ensure a fair comparison, we decrease the training epochs (outer loop) in order to guarantee that ASM has same number of back-propagation in each experiment. The results are reported in Fig. 1(a) and (b). AS we can see, the best depth choices for cross-domain classification and segmentation tasks are $n = 2$ and $n = 5$, respectively.

Figure 1: **a:** Classification performance in terms of accuracy when using different search depth. **b:** Segmentation performance in terms of mIoU when using different search depth. **c:** A sweep of performance over varying fraction of unlabeled target samples.

Leveraging a very large or very small depth would do harm to the performance for both tasks. On the one hand, when using $n = 1$, ASM would degrade into a baseline that uses only anchor styles during the training process, which deactivates the *DE facto* adversarial paradigm. One the other hand, a very large search depth would lead $G$ to generate many unreasonable styles that are excessively deviated from the target distribution. Based on this variant study, we respectively choose $n = 2$ and $n = 5$ for classification and segmentation tasks.

**Variant Study on Different Amount of Target Sample.** Although the proposed ASM mainly aims at OSUDA problem, we evaluate it under zero-shot UDA (ZSUDA) and few-shot UDA (FSUDA) setting to evaluate its robustness in this experiment, where less or more than one samples are available during the adversarial training. Specifically, the percentage of the unlabeled target sample varies over a range {0, 25, 50, 100}. Note that the anchor style for ASM is initialized randomly under the ZSUDA setting. As we can observe in Fig. 1 (c), on the one hand, ASM outperforms OST in all ZSUDA or FSUDA cases. On the other hand, ASM can bring a huge boost to the accuracy comparing to ZSUDA when given a sole sample, indicating that ASM is able to capitalize on the sole target sample for domain adaptation. Finally, we find that increasing the sample number is still a very effective way to improve classification accuracy. Such performance gap between OSUDA and FSUDA provides both opportunities and challenges for the future research on OSUDA problem.

**More Visualization Results of the Searched Stylized Images.** We show more visualization results of the searched stylized images by ASM in Fig. 2 and Fig. 3, respectively.

Figure 2: Visualization of the searched stylized image sequence by ASM in cross-domain classification task.

Source Images      Sole Target Image      Anchor-Style Images      Searched Stylized Images by ASM (search depth = 4)

Figure 3: Visualization of the searched stylized image sequence by ASM in cross-domain segmentation task.

## Footnotes

\*Corresponding author (qd_gt@hust.edu.cn).