[Reviews · NeurIPS 2020]

Review 1

Summary and Contributions: This paper aims at solving One-Shot Unsupervised Domain Adaptation, which assumes that one target labeled sample is available during adaptation. The atuhors propose Adversarial Style Mining approach to consider iteratively style transfer and learning from transferred images to train the adapted model for target domain. Their proposed ASM introduces the adversarial training paradigm which can benefit specific downstream task. The method achieves state-of-the-art adaptation performance using this method on one-shot domain adaptation setting.

Strengths: The proposed ASM investigates a novel way from style transfer perspective to do domain adaptation. This work is different from previous works (e.g. CycleGAN, Cycada). It uses differentiable AdaIN to determine what’s the beneficial style statistics for specific downstream task (e.g. semantic segmentation) and target datasets (e.g. cityscapes). The analysis and experiment setup of Random AdaIN are very detailed, which fully explains how the model gradually learns the correct style statistics. Moreover, the method achieves state-of-the-art performance and has an instrumental effect in simulation to real adaptation area.

Weaknesses: In general, my concerns about this paper mainly include two points. About Random AdaIN (RAIN) In this paper, the author emphasizes the module RAIN, and we can see that this module is actually one of the main contributions here. But it just looks like a more complex differentiable version AdaIN. Differentiable version is indeed a good thing, which can help to find a better way for style transfer to specific tasks and datasets iteratively. But I don’t find the ablation study between RAIN and AdaIN. To my understanding, AdaIN can work in UDA setting logistically. If RAIN is just a complex improvement on Semi-DA or One-Shot DA of AdaIN, I don’t think this part is very attractive because it also needs special structure and more computations. About Style Transfer For the image classification experiment, I don’t think it’s persuasive enough, because it might be useful when you have some similar domains like Digits-5. But for other datasets like VisDA or DomainNet, style transfer methods won’t work well. Therefore, I believe that style transfer method is only applicable to semantic segmentation/object detection tasks, especially on the datasets which train on GTA and test on Cityscapes. I don’t think this is a big problem for this paper, but in fact, as the paper focuses on it, it will make people feel that it is an improvement on a certain situation rather than a substantial promotion of domain adaptation or transfer learning. Therefore, I do not recommend that NeurIPS should help to credit such work.

Correctness: The ASM includes RAIN and FCN parts. I believe their coordination is correct.

Clarity: First of all, this paper describes the design of RAIN in detail, and illustrates the role of RAIN through experiments and diagrams. Secondly, the mathematical symbols of this paper are all correct.

Relation to Prior Work: This paper does compare with ASM with some most recent work on this direction. In terms of performance, it really exceeds previous works. In addition, citations for current style transfer works are also relative adequate. But the only thing that concerned me is that I don’t find any comparision with the other CycleGAN-based methods, because to my understanding, these parts of works can generate more stylized transferred images without supervision than AdaIN. It seems there’s no comparisons or discussions with CycleGAN-based style transfer methods. At least, from the figures generated by ASM in the supplementary material, they don’t look very close to the target domain.

Reproducibility: Yes

Additional Feedback: From the perspective of a researcher, I appreciate the author's work, which provides a good idea for reference for Domain Adaptation field. However, from the perspective of a NeurIPS reviewer, I do not think this high-level conference should encourage a relatively minor revised work. After rebuttal: Thank the authors for replying to my concerns. After going through other reviews and the rebuttal, the authors have addressed some concerns. However, I still think the algorithm is not theoretically principled and the improvement is marginal. So I will keep my negative score but upgrade to 5.


Review 2

Summary and Contributions: In this paper, authors propose the Adversarial Style Mining (ASM) model for one-shot unsupervised domain adaptation. In specific, a style VAE is pretrained to generated different target-like images based on the latent code sampled from a Gaussian distribution determined by the single target sample. In the training process, the style transfer module and a task-specific module is co-trained in an adversarial manner, such that the task-specific model is gradually enhanced by more and more difficult stylized samples. The experiments on digital classification and synthetic to real segmentation tasks show that the proposed model achieves superior performance.

Strengths: + The use of style VAE for image transformation from source to target domain is technically sound. + The co-training of style transfer and task-specific module in an adversarial manner is novel. + As a whole, the proposed ASM model is well suited for one-shot unsupervised domain adaptation.

Weaknesses: - I have only one concern whether the proposed model can equally perform well on those domain adaptation tasks where the shift of style between two domains is more obscure, e.g. the classification task on Office-31 dataset and the segmentation task adapting from KITTI to Cityscapes dataset.

Correctness: The proposed claims and methods are conceptually and technically sound.

Clarity: The paper is well written that I cannot find major flaws in it.

Relation to Prior Work: This work extensively discusses the relations and differences between previous works in style transfer and (one-shot) UDA fields.

Reproducibility: Yes

Additional Feedback: Questions: 1. Whether the proposed ASM model can also perform well on those domain adaptation tasks where the shift of style between two domains is more obscure? It would be more convincing to also evaluate on Office-31 dataset and the segmentation task adapting from KITTI to Cityscapes dataset. Suggestions: 1. In the broader impact section, it would be better to mention the underlying risks of this work. After the author response period: After reading other reviews and authors' feedback, I confirm my initial position that this work is a good attempt to tackle the One-Shot UDA, especially under the scenarios with style-induced domain shift. I'll keep my positive rating.


Review 3

Summary and Contributions: This paper proposed a new problem named one-shot unsupervised domain adaptation and a new method to solve it by generating various stylized images to strengthen the segmentation model.

Strengths: The proposed method to generate images with different styles is interesting and help train a stronger segmentation model, which is even better than a lot of UDA models that are trained with more real images. The experiments in the main paper as well as the supplemental material are well designed.

Weaknesses: 1. The motivation of this work is very unclear. The author assume both labelling and collecting data are not trivial. I can understand labeling a large image for segmentation task is not easy. But there are millions of pixels in one image. But I don't see any reason that collecting images is hard as well. The author does not give any explanation about it. As the author works on a new problem, it is quite crucial to me that the motivation to do it makes sense, otherwise why we consider this problem. 2. It is not quite clear to me how the single target image is used. After I read the paper especially Figure 2 and 3, I don't find any clear description about how to use the target sample. The author mentioned that they use the target sample as an anchor image to generate similar stylized images. But it is not clear to me how to achieve it. It seems to me that the author just use Gaussian Random noise to generate new images and feedback from the loss is used to guide how to generate noise.

Correctness: Yes

Clarity: Yes

Relation to Prior Work: Yes

Reproducibility: No

Additional Feedback: Please refer to the weakness. My question is answered well in the rebuttal. So I choose to move the score to positive.


Review 4

Summary and Contributions: This paper proposed to tackle a new problem scenario: one-shot unsupervised domain adaptation, where there is only a single unlabelled target-domain sample available when learning to adapt from the source to the target domain. The proposed method is stemmed from the idea of performing stylization on sources images to mimic the target-domain samples. With learning a latent space of style features in the stylization module, the stylization module and the task model (e.g. segmentation or classification) becomes the adversarial pair, where the task model should try to well recognition the stylized source images while the stylization module would search for (in the latent space of style features) harder stylization to trick the task model (named as adversarial style mining in this paper), thus improving the capacity of the task model on performing recognition on target-domain samples during the testing time.

Strengths: + The problem of one-shot unsupervised domain adaptation is relatively new and also challenging. The idea of having adversarial pair between stylization module and task model to boost the domain adaptation is also novel. The experimental results demonstrate the superiority of the proposed method with respect to several domain adaptation baselines, in particular the proposed method, by using only a single target-domain sample, is able to provide competitive performance in comparison to other baselines of utilizing rich target-domain data.

Weaknesses: - As the problem scenario is actually inbetween the typical unsupervised domain adaption and the domain generalization, adding some discussion or even comparison with respect to domain generalization approaches could further enhance this submission. - The performance of the proposed method is actually quite sensitive to the setting of the search depth (i.e. how many new stylizations other than the style given by the single target-domain space are found during the adversarial style mining) as well as the sampling strategies (e.g. stylization based on random samples in the style latent space or based on the styles found during the adversarial style mining, in which for the latter it is initialized from the style extracted from the given target-domain samples). It would be actually interesting to discuss whether there can be a rule of thumb to determine the search depth, as well as checking if the combination of different sampling strategies could be beneficial for the domain adaptation. - In terms of motivation, it would be also better to give some examples of the practical applications where such one-shot domain adaptation is truly needed. - The proposed method is based on the assumption that the distribution gap between source and target domain is mainly caused on the style difference, in which it is actually limited but still reasonable as this work is one of the first attempts to address a challenging problem scenario. Any ideas to release such assumption would be great to have in the paper. - As the adversarial style mining highly depends on the initial style from the given target-domain sample, it would be better to see how different one-shot target-domain samples result in domain adaptation performance. For instance, if now the one-shot target-domain sample is taken from a night scene (quite dark in most of regions), how well does the task model after domain adaptation handle day scene (and vice versa)?

Correctness: The claims and the proposed method in this paper seem to be correct. The only concern is that the latent space of the style features could potentially be not that smooth to search for the reasonable stylization in the procedure of adversarial style mining.

Clarity: Yes, this paper is well written and easy to follow.

Relation to Prior Work: Yes, the problem scenario of this paper clearly differs from the prior works of typical unsupervised domain adaptation.

Reproducibility: Yes

Additional Feedback: Overall I found this submission interesting and novel. It would be great if the authors could provide response to my comments for the Weaknesses and Correctness. After rebuttal: I have thoroughly gone through all the reviews and the authors' feedback, where I found quite some common concerns shared among the reviewers (e.g. the practical scenario of such one-shot unsupervised domain adaptation setting, missing discussions related to the domain generalization or domain randomization, discussion on dealing with the domain shift other than style difference) and the authors do provide reasonable responses in the rebuttal. Basically, I am still positive about this submission and willing to see it being accepted to the NeurIPS2020. However, I would encourage the authors to provide in their final camera ready more detailed discussion on the practical usage of the proposed scenario (even to have some showcases, e.g. rare disease information with privacy or videos under extreme weather conditions as mentioned in the rebuttal), the connection to the domain generalization or domain randomization (e.g. to add the comparison against the baseline of adopting data augmentation, as pointed out by R5), and the investigation on the limitation of the proposed method (e.g. some failure cases).


Review 5

Summary and Contributions: This paper introduces a method for unsupervised domain adaptation, where the training inputs are stylized to look like the test inputs. A new stylization method, RAIN, is introduced to learn a variational distribution over AdaIN parameters. This distribution, conditioned on the test style, is then sampled from to stylize the training data to look like the test data, but with random variation. This sampling can be performed adversarially so that the training data is stylized in a way that is as hard as possible for a recognition netowrk to correctly label, referred to as ASM. Results are showcased primarily on the one-shot setting but the method can also be adapted to zero shot, few shot, etc settings.

Strengths: + Novel set of ideas + Method intuitively makes sense + Decent results + Reasonable set of experiments to validate and ablate the method

Weaknesses: - Method is rather complicated and its hard to be sure where performance is coming from (even with the ablations), or to extract clear lessons - Writing is a bit hard to follow - Theoretical grounding / underlying principles for why this works are unclear - Missing discussion of domain randomization / data augmentation methods

Correctness: I don't see any errors in the methodology, although not all details are easy to follow. The model itself feels a bit ad hoc and it's unclear why, theoretically, it should work.

Clarity: The clarity could be improved. Mostly at the sentence level: there is some clunkiness, typos, word choice and grammatical issues, etc. Effort to simplify the exposition would be appreciated.

Relation to Prior Work: The prior work on domain adaptation is adequately discussed and compared against. However, the paper does not mention the large body of work on domain randomization as an alternative to domain adaptation, e.g., [1,2]. Arguably, the proposed method is closer to domain randomization, where training data is randomly augmented to achieve robustness to stylistic changes in the test set. The adversarial training scheme is reminiscent of "automatic domain randomization" [3]. These methods should be discussed, although there is not a direct way to compare against them since domain randomization methods typically make use of a simulation engine. What could be directly compared against is simply using data augmentation. I'd be curious to see the results of just using standard data augmentations, like cropping and color jitter, as the way to "stylize" the training data. Recent work (e.g., [4]) has also tried using generative models for the data augmentation, which starts to look closer to the kinds of stylizations in the current paper. This literature should at least be discussed. [1] https://arxiv.org/abs/1703.06907 [2] https://arxiv.org/abs/1611.04201 [3] https://arxiv.org/abs/1910.07113 [4] https://arxiv.org/abs/1711.04340

Reproducibility: Yes

Additional Feedback: I found this to be an interesting paper but it was hard to pinpoint what was driving performance, because there are a lot of moving pieces and details to get right. A few questions and concerns: Are the comparisons to baselines, like OST, fair? Were the neural net architectures the same? Were they trained for the same number of epochs? etc? I realize it's impossible to perfectly equate such highly disparate methods. However, the performance gains are rather small and to me it's not fully convincing that the new ideas in this paper are responsible for the performance gains. The ablation studies are more interesting and convincing to me. Some of the most interesting results are in Section C of the appendix. I'd suggest moving that to the main text, in particular the plots showing performance as a function of unlabeled data and the search depth plots. While reading the method, I was left wondering: why doesn't the adversary make up an impossible style, like turning the image completely to black? This failure mode should be discussed in the method section, otherwise the adversarial training comes across as unprincipled and potentially degenerate. The search depth plots seem to show that this kind of failure can occur and the trick is just not to train the adversarial too much. I found the focus on the one-shot setting to be somewhat contrived. Is this really an important practical setting? Some additional motivation would be useful. It may be easier to motivate few-shot learning, rather than overly emphasizing _one_ shot. Few-shot can encompass not just one-shot but also zero-shot (which is what domain randomization deals with) and k-shot, which is perhaps the more practical setting (usually we can get at least a few images from the test domain). The results in the appendix seem to indicate that the proposed method is effective in all these settings, so why focus the story only on one-shot? One missing experiment I would like to see is replacing ASM with just random sampling about the achor. The sampling radius could be tuned as a hyperparameter. The adversarial scheme seems overly complicated to me, and prone to degeneracy. I wonder if something simpler and safer would do just as well. ------ Post-rebuttal update ------ Thank you for the clarifications in the rebuttal, especially regarding the comparison to random sampling, which I had missed. I remain positive about the paper and have therefore not changed my score. Discussing domain randomization and data augmentation will be very useful I think.

[Author Response · NeurIPS 2020]

We thank the reviewers for their thoughtful feedback! We are encouraged they found our idea to be novel (R1,R2,R4,R5),
our performance remarkable (R1,R2,R3,R4,R5) and identified our contribution to this challenging OSUDA problem
(R2,R3,R4,R5). We are pleased to get a positive average score where R2,R4 and R5 gave positive feedback. We begin
by answering two **common concerns (CC)**. We will incorporate all feedback in the revision.

**R3,R4/CC1.** About the motivation. **A.** Here we'd like to emphasize our motivation for ASM again. Our core viewpoint
is that not only data labeling but also data collection itself might be challenging. Such challenge could come from data
privacy or acquisition condition. For example, it could be hard to acquire rare disease information with privacy or to
shoot videos under extreme weather conditions. In fact, we have given some examples in **Broader Impact** for it. We
believe our spirit is in line with the prevailing trends of few-shot learning.

**R1,R2,R4/CC2.** When facing more obscure domain shift beyond "style difference". **A.** We thank reviewers for the
insightful concern. Style difference is one of the vital causes for domain shift, while ASM is tailored for addressing
such style gap. Howbeit, we acknowledge that ASM is an early attempt towards the challenging OSUDA problem
and has it own range of application. As we pointed out in **Conclusion**, we left it as the future work to cope with more
general domain shift such as Office-31.

**R1/Q1.** RAIN seems only a complex version of AdaIN, which is not very attractive. **A.** We thank R1 for identifying
the novelty and superiority of our work. However, maybe we did not illustrate it clear enough and make R1 miss the
focus of our contribution. The uppermost contribution of our work is the design of an adversarial paradigm (ASM)
tailored for OSUDA, as acknowledged by other reviewers, while RAIN should only be regarded as an indispensable
module to achieve the paradigm. Besides, we argue that RAIN is not a "mere complex", but a *premium* version of
AdaIN because:1) RAIN has benefits of end-to-end training and differentiable searching with negligible computation
cost; 2) We can regard "anchored sampling" (see **Appendix C**) as AdaIN in OSUDA scenario. As demonstrated in this
ablation study, the new style generated by AdaIN is much limited comparing to RAIN, especially in a one-shot setting.

**R1/Q2.** No comparison with CycleGAN-based methods. **A.** In fact we have compared ASM with CycleGAN directly
in both classification and segmentation task (see **Table 1 and 2**). Besides, we have compared ASM with OST and
MUNIT, which are both CycleGAN-based methods.

**R3/Q1.** How to use the single target sample? **A.** We have given a detailed description on how the single target sample
is used in **Figure 3** and **Algorithm 1**. Please allow us to use simple sentences here to explain it again. The initial style
vector $\varepsilon_1$ is indeed from a Gaussian distribution, but such Gaussian distribution is defined by $\psi$ (mean) and $\xi$ (variance).
**$\psi$ and $\xi$ are both extracted from the single target sample $x_t$ by RAIN.**

**R4/Q1.** Lack of discussion on domain generalization. **A.** Different from OSUDA setting, most DG methods leverage
multiple labeled source domains but no target data. We will add discussions on these methods in revision.

**R4/Q2.** Sensibility to the search depth. **A.** The performance is not very sensitive to search depth if the depth is in an
appropriate range. For classification, the accuracy drops around 2% when depth $5 \rightarrow 10$. For segmentation, the mIoU
drops around 1% when depth $2 \rightarrow 4$ (see **Appendix C, Fig.1**). An overlarge search depth would lead to unreasonable
styles, so it is easy to determine an appropriate depth by observing the generated samples.

**R4/Q3.** Sensibility to the target sample choice. **A.** We think it is a common issue for the one-shot learning, and our
answer is twofold. 1) ASM has a wide search scope, as shown in **Fig.4 (Right)**. Taking the day and night scene as an
example, when ASM learns the dark style well, the search direction may change to the bright style. The wider search
space relieves the performance sensitivity to the target sample. Besides Fig.4, We will give more visualisation analysis
in revision. 2) We run each OSUDA experiment for 5 times with different target samples (see **Sec. 4.1**). We find the
performances are stable in most cases. In summary, we do not need a specific target sample for good performance.

**R4/Q4.** Latent space smoothness. **A.** We use a **large** dataset of style images to train RAIN. According to the VAE,
all these reasonable styles are embedded in a Gaussian distribution and the latent space is supposed to be dense and
smooth. As we observed in experiments, the styles searched around the anchor are reasonable.

**R5/Q1.** It is hard to be sure what drives the performance. **A.** We experimentally proved that ASM drives the performance.
RAIN alone (anchored or random sampling) performs equally with OST (see Tab.2 in main text and Tab.2 in appendix)
but ASM consistently outperforms OST. It indicates that ASM mechanism is the promoting factor while RAIN is only a
module to achieve the mechanism. Besides, we believe the comparisons are fair. We uniformly use ResNet-101 for seg.
and ResNet-18 for clas., using same data augmentation and same number of training epochs.

**R5/Q2.** Writing could be improved and simplified. **A.** We will polish our writing to make it easier to follow.

**R5/Q3.** About theoretical grounding principle. **A.** ASM employs the searched target styles to stylize the source images
in order to decrease the domain distribution discrepancy in input space, which is consistent with the theory of Ben-David
et al [1]. Besides, the establishment of the adversarial mechanism is inspired by the the Gradient Reverse Layer (GRL)
[2]. We will add these theoretical insights in revision. **R5/Q4.** Missing discussion. **A.** We will add detailed discussions
and cite related papers about domain randomization and data augmentation. **R5/Q5.** Unreasonable styles. **A.** Please
kindly refer to R4/Q2 & R4/Q4. //**[1]** A theory of learning from different domains. Machine Learning 2010. **[2]**
Unsupervised Domain Adaptation by Backpropagation. ICML 2015.

[Meta-Review · NeurIPS 2020]

This paper tackles the recent problem of one-shot unsupervised domain adaptation and proposes a new model called Adversarial Style Mining (ASM) to solve it aimed at generating various stylized images for the benefit of downstream tasks (segmentation or classification). After the rebuttal, there is a large consensus that the work has some interesting aspects such as novelty, technical soundness, motivations and valuable results. It was missing a more thourogh discussion about the several steps of the method and comparisons wrt other related approaches (e.g., domain randomization), which seem to be partially addressed by the rebuttal. In the final version, authors are encouraged to address more carefully the reviewers' final remarks and suggestions.